# The Interaction Effect of Type of Message X YouTuber's Media Metrics on Customers' Responses and the Moderation of Conformity Intention

**Melby Karina Zuniga Huertas [1,\*] and Tarcisio Duarte Coelho [1,2]**

[1] Department of Management, Centro Universitário da Fei, São Paulo 09850-901, Brazil; tarcisiocoelho@gmail.com
[2] Department of Management, Toledo Prudente Centro Universitário, Presidente Prudente, São Paulo 19030-000, Brazil
\* Correspondence: mhuertas@fei.edu.br; Tel.: +55-11-3207-6800

**Abstract:** This is a study of the way in which YouTubers' media metrics influence the effect of their one-sided messages (1SMs) and two-sided messages (2SMs), providing theoretical explanations based on the elaboration likelihood model. Its main objective is the proposition and testing of: (i) the interaction effect between type of message and media metrics of the YouTuber on customers' responses, and (ii) the moderation of individuals' conformity intention for the interaction effect between type of message and media metrics on customers' responses. The results of an experiment showed that high YouTubers' media metrics have more effect for 1SMs and less effect for 2SMs. Additionally, conformity intention moderates the effect of the interaction type of message X media metrics. A high level of conformity intention neutralizes the interaction effect between YouTubers' media metrics and message sidedness. This study makes a theoretical contribution to research into online content and information use, providing explanations of how media metrics of a vlog influence the effect of two types of messages.

**Keywords:** two-sided messages; one-sided messages; YouTube; social network; conformity intention; media metrics

---

## 1. Introduction

This is a study of the way in which YouTubers' media metrics influence the effect of their one-sided messages (1SMs) and two-sided messages (2SMs), providing theoretical explanations based on the elaboration likelihood model (ELM). An effective way of communicating persuasively is by using 2SMs, in which the emitter describes both sides (i.e., positive and negative) of an issue but actually still favors one side [1]. Conversely, 1SMs present only positive characteristics [2]. Much studied in traditional media advertising, 2SMs inform the positive primary characteristics of an offer as well as at least one negative secondary characteristic, which strengthens perceptions of source credibility [3–9], reduces negative cognitive responses, and has a positive effect on brand attitudes and purchase intention. The use of 2SMs has increased, and customers are no longer limited to traditional, one-way, seller-to-buyer communications [10] but are now exposed to more sources of information, such as network online communities [11], and different kinds of information, such as product (goods and services) experiences [12] and revisions [13]. This means that customers are exposed to messages about products in many different ways. For example, consumers could read an editorial about the pros and cons of buying a specific product, watch a YouTuber evaluating its functionalities, and join a related online community. Those customers would then be aware of both positive and negative

product characteristics, with information provided by independent or commercial sources that would influence their responses towards the product and the brand.

The literature review identified a strand of theoretical development in the study of the effect of 2SMs, namely to establish the limits of the effect of 2SMs by analyzing how the effect occurs in different situations (e.g., online environments such as social networks and websites). In this regard, there have been several attempts to model and interpret the effect of 2SMs [3–9,14–18]. Nonetheless, the heterogeneity of the results, as pointed out by [9], still indicates the need for further investigation of substantive moderating variables [16]. In this research, it is proposed that the effect of 2SMs in online environments could be different from their effect in traditional advertising media. In contrast to one-way distribution of information, social media engage users in two-way communication [19]. Thus, in online environments, it is known that compliance with social norms can emerge in different ways compared to those observed in face-to-face interaction [20]. However, it is still unclear which elements have the power to influence individuals' behavior during online communication processes [2], calling for more specific explanations.

YouTube, initially a purely user-generated content platform, has evolved to a platform for professionally produced content, showing that its content has become an important marketing tool [21]. There are three types of content on the Internet [22]: content sponsored by a company, a brand or a product, whose objective is to persuade customers (e.g., a YouTuber advertisement of a product); content marketing, focusing on particular interests of customers and free of sales messages (e.g., a Facebook brand page); and user-generated content, directed at help or entertainment (e.g., product reviews and shopping tips) [22]. This research focuses on user-generated content on YouTube delivered by video blogs (known as 'vlogs' or 'vlogging' carried out by 'vloggers') [23]. This type of content is highly relevant to the marketing strategy of a company, often leading it to offer vloggers free products or services, gift cards, or money. In exchange, the vloggers provide positive and interesting content about the company's products or services in their vlogs [24]. Nevertheless, the effect of YouTubers' messages on customers' responses has yet to be explained, reinforcing the relevance of this research. Online content providers have increasingly incorporated technological tools, designed to allow users to express their opinions about specific media content and related issues, generating social media metrics. Social media metrics often provide cues about other users' media exposure, as well as their implicit attitudes and behaviors [25] by disclosing numbers of likes, shares, subscribers, comments, etc. Typically displayed alongside each content in social networks, social media metrics provide indications of the popularity or virality of a specific content [26]. However, different from its effect on virality, little is known about how social media metrics shape the way people estimate the influence of different types of media content on the self [27]. In this context, it is relevant to analyze the effects of social media metrics and the specifics of content on individuals' responses.

Previous studies have shown that the effect of 2SMs depends upon situational factors, such as: disclosure uniqueness, cognitive load and involvement [9], time pressure [14], and the importance of the negative attributes [18]. There is also evidence that the effect of the message may be moderated by some of its characteristics, such as regulatory congruence [15], issue ambivalence, and message arguments [16]. Similarly, the impact of 2SMs may be moderated by individuals' characteristics, such as self-regulatory focus [28], cognitive effort [9], and the need for cognition [14]. This research starts with the premise that although 2SMs generate more credibility, improving results in traditional media, their effect on customers' responses will be different in social networks. Specifically, the interaction effect of media metrics and type of the message (1SM and 2SM) on customers' responses is proposed. In this proposition, we are including a new explanatory factor to the sided message's effect literature. Additionally, as confirmed in research regarding content media virality, the moderation of the individual's conformity intention on the already stated interaction effect is proposed. Both propositions of moderation aim to provide explanations on how the effect of 1SMs and 2SMs occurs in online environments.

Focusing on YouTubers' messages about a product, the main objective of this study is the proposition and testing of: (i) the interaction effect between type of message (i.e., 1SM and 2SM) and media metrics of the vlog (i.e., the number of subscribers of the YouTuber and the number of likes and shares of the vlog) on the customer's responses; and (ii) the individual's conformity intention (regarding their behavior in social networks) as a moderator factor of the interaction effect between the type of message and media metrics of the vlog on the customer's responses. This research supports the proposed moderators' effects explanations and its hypothesis mainly through the ELM.

## 2. Hypotheses Development

There are surface characteristics and features in online environments that are not present in traditional media (e.g., information about website visitors, video likes, and Facebook page friends) that influence the way in which a message is processed and its consequent effects on customers' responses. For example, during the processing of a print advertisement, the customer does not have information about how many people have seen the ad. Previous research has demonstrated that mass audiences typically exhibit a propensity to gravitate toward content that has already established some popularity, because individuals associate the quantity of views with the quality of its appeal [29]. Internet users rely mostly on surface characteristics and features (e.g., media metrics, appearance) rather than on independent, content-based evaluation to make judgments about the credibility and quality of information [30]. This would happen because of the users' superficial approach to assessing materials and means that they are susceptible to their perceptions of peer preferences or opinions that are on display [30]. The authors conclude that user choices tend to be successively imitative in nature, and that the ongoing audience capture of videos depends on their previously acquired viewer bases, manifested by view counters. They support their explanations by citing the 'Bandwagon Theory' [31], focusing on the effect of previous viewer bases on customers' views and shares of viral videos. However, the study did not look at the specifics of content, which also may matter when attracting viewers [30] and advertising products. By contrast, in the current research, we extend the examination of how a surface characteristic of the vlog (i.e., a media metrics), and the type of message by which a product is presented (i.e., 2SM or 1SM) affects customers' attitudes towards the vlog, the product, and the brand. The media metric under examination in this research is information about a YouTuber's numbers of subscribers, likes, and shares.

Previous research explains the effect of media metrics on individuals' responses by the heuristic–systematic model [32]. It defines the individual's reliance on mental shortcuts as heuristic processing and suggests that individuals tend to choose this route unless they are highly motivated to process issue-relevant information [33]. Particularly, media metrics can trigger a bandwagon heuristic, which refers to the judgment rule in which individuals base their perceptions and behaviors on other people's reactions [32]. This process is explained by the ELM, which posits that there are two routes to attitude change depending on cognitive preconditions [34]. When there is low motivation and ability to process the message, people tend to follow the peripheral route, relying on simple cues to make a decision without complex cognitive efforts. In the bandwagon heuristics, the individual is following the peripheral route. By contrast, when there is high motivation and ability to process a message, persuasion is likely to occur through the central route, a careful and elaborated processing of information. This would be the case of 2SMs, because they will increase attention and motivation by augmenting source credibility of the message, leading to cognitive responses, perceived novelty, and positive attitude towards the brand, the last one producing an effect on purchase intention [9]. This means that after exposition to a 2SM, people tend to follow the central route of the ELM.

We support our reasoning with the ELM. Consumers evaluating a 1SM, by following mostly the peripheral route, will be more likely to neglect the value of the information and rely merely on the number of attributes and the peripheral features as media metrics. Under this condition, individuals would have better responses when high media metrics were disclosed than if there were no information about media metrics. On the other side, individuals evaluating the arguments of a 2SM mostly by

the central route will be influenced by the negativity of the information provided in the message, increasing credibility and raising responses towards the message. Consequently, the hypotheses for testing are:

–　　H1a: When exposed to a 1SM vlog of a product informing about a high YouTuber's media metrics, an individual will have a more positive attitude towards the vlog than if exposed to a vlog without information about the YouTuber's media metrics.

–　　H2a: When exposed to a 1SM vlog of a product informing about a high YouTuber's media metrics, an individual will have a more positive attitude towards the product than if exposed to a vlog without information about the YouTuber's media metrics.

–　　H3a: When exposed to a 1SM vlog of a product informing about a high YouTuber's media metrics, an individual will have a more positive attitude towards the brand than if exposed to a vlog without information about the YouTuber's media metrics.

On the other hand, a 2SM vlog emitted by a YouTuber with high numbers will generate worse responses towards the vlog than if no information about YouTuber's media metrics were given. The 2SM, including a negative characteristic of the advertised product, would enhance attention and motivation to process the message [9]. For this reason, a vlog without information about a YouTuber's media metrics would be perceived as a neutral opinion of the YouTuber, increasing credibility and raising responses towards the message [9]. The individual will perceive their individuality since there is not information about others' opinions and will maintain their individuality, processing the 2SM by the central route. By contrast, when high media metrics are disclosed, the individual will be more critical of the 2SM, which would be perceived as a disguised manipulation attempt of advertisement. Our reasoning is that negative information about the product may activate some initial disagreement towards the message and for this reason, the individual will tend to engage in counter-arguing. Consequently, they will decrease their attitude towards the vlog with high media metrics. Thus, when exposed to a vlog with a high YouTuber's media metrics, an individual will have a less positive attitude towards the vlog, the product, and the brand than towards a 2SM vlog without information about the YouTuber's media metrics. Consequently, the hypotheses for testing are:

–　　H1b: When exposed to a 2SM vlog of a product informing about a high YouTuber's media metrics, an individual will have a less positive attitude towards the vlog than if exposed to a vlog without information about the YouTuber's media metrics.

–　　H2b: When exposed to a 2SM vlog of a product informing about a high YouTuber's media metrics, an individual will have a less positive attitude towards the product than if exposed to a vlog without information about the YouTuber's media metrics.

–　　H3b: When exposed to a 2SM vlog of a product informing about a high YouTuber's media metrics, an individual will have a less positive attitude towards the brand than if exposed to a vlog without information about the YouTuber's media metrics.

In online environments, individuals' cues (e.g., anonymity, arousal, and sensory overload) could affect their interaction with other participants, as well as their responses towards a message. In this regard, deindividuation theory explains how circumstances occurring in group situations (e.g., anonymity, loss of individual responsibility, arousal, sensory overload and unstructured situations) result in a deindividuated state, which is antecedent to antinormative and disinhibited behavior [35]. A deindividuated state is indeed likely to occur in online environments, and under specific conditions, it can become a powerful tool to trigger conformity among individuals. The explanation for this conformity is that while deindividuated, subjects have a diminished perception of their personal traits, and so, if the group the subjects are interacting with is made salient, then the subjects will be more likely to conform [36]. In other words, group identification is positively associated with conformity intention, while perceived deindividuation makes individuals reluctant to agree with others [37]. Thus, theoretically, the individual's identification with the group mediates the effect of deindividuation on

conformity intention. This means that a deindividuated individual, who is highly identified with the group, will increase their intention to conform to the group norms. By contrast, a deindividuated individual, who is not identified with the group, will reduce their intention to conform. In the studies mentioned previously, conformity intention to group norms was analyzed as a dependent variable. In a broad perspective, conformity intention to group norms is the consequence of a deindividuated state when the group is salient. However, in a more restricted perspective, could an individual's conformity intention to a group's opinions in social networks interactions influence their behavior? Oriented by this inquiry, and differing from previous research, in this article, we propose a new perspective for the analysis of conformity intention. We focus on conformity intention as the individual's intention to conform to the group opinions when they are interacting in social networks. It is known that conformity, also called herding behavior, is one of the influences of being in a social network [38]. Conformity refers to the person's inclination to be influenced by others' opinions [37]. In a broader perspective, conformity is defined as the tendency of changing behavior in order to fit in a group and converge to the group norms [39]. In this research, we define conformity intention as the respondents' intentions to agree with any dominant opinion promoted by others in social networks. It is argued that conformity intention related to social network behavior could moderate the interaction effect between type of message (1SM and 2SM) and a YouTuber's media metrics. In addition, the proposition and testing of H1, H2, and H3, we add to the analysis the moderation of the individual's level of conformity intention towards a social network's effect. Consequently, the Hypotheses to be tested are:

- H4: The level of an individual's conformity intention moderates the effect of the interaction between type of message and a YouTuber's media metrics on attitude towards the vlog.
- H5: The level of an individual's conformity intention moderates the effect of the interaction between type of message and a YouTuber's media metrics on attitude towards the product.
- H6: The level of an individual's conformity intention moderates the effect of the interaction between type of message and a YouTuber's media metrics on attitude towards the brand.

The proposed hypotheses, graphically represented in Figure 1, were tested empirically using an experimental approach.

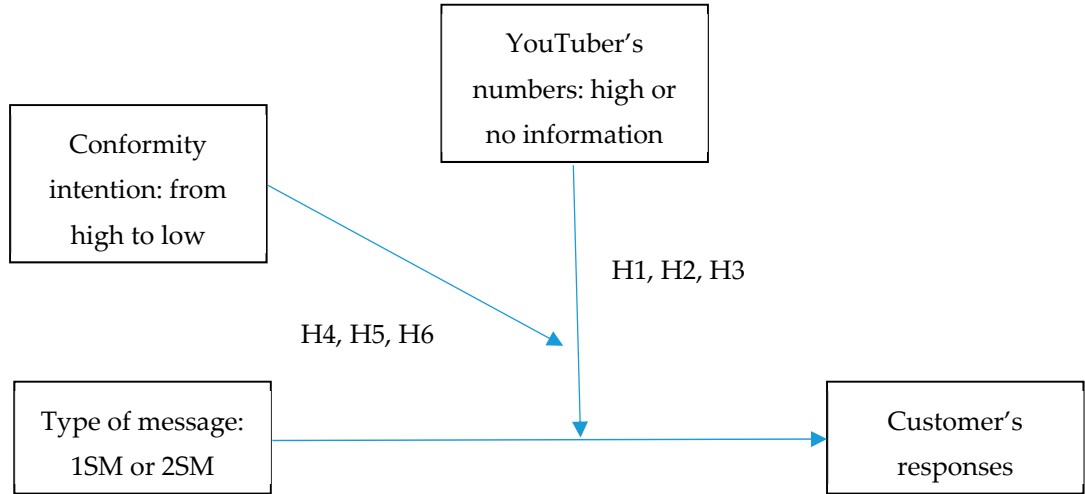

**Figure 1.** Proposed theoretical model for the interaction effect of type of message X YouTuber's media metrics and the moderation of conformity intention.

## 3. Experimental Study

The experimental study was designed to test: (i) the effect of the interaction between type of message X YouTuber's media metrics on customers' responses, stated in H1, H2, and H3; and (ii) the moderator effect of conformity intention on the effect of the interaction type of message X YouTuber's

media metrics on customers' responses, stated in H4, H5, and H6. A $2 \times 2$ factorial design (type of message: 1SM vs. 2SM, YouTuber's media metrics: high vs. no information) was used.

### 3.1. Independent Variables Conditions

To determine conditions for the variable type of message, a first focus group was held to discuss salient topics and products for young people. The focus group consisted of 12 undergraduate students at a business administration school, 6 of whom were female and 8 were between 21 and 25 years old. 'Tips for a healthy life' was identified by the participants as a salient topic among young people. Based on these results, the product category selected for this experiment was a supplement for healthy nutrition. Then, a second focus group with the same 12 students was held to discuss both the primary and secondary, and positive and negative characteristics of healthy nutrition products. Regarding these results, the researchers manipulated the type of message conditions. Two vlogs were produced for a new nutrition supplement, one containing a 1SM and the other a 2SM. They starred the (fictitious) YouTuber Renata Duarte and were recorded using resources common among YouTubers. In both vlogs, the title was 'Tips for a heathy life'. The YouTuber begins by discussing the importance of being healthy, having the right nutrition, and practicing sports. Next, she talks about some things she has started to do that she feels are having good results for her: (i) drinking more water; (ii) working out with a personal trainer; and iii) having healthy nutrition. When she talks about healthy nutrition, she discusses a product that has helped her a lot (a protein chocolate fudge named Power Two). When she is talking about the product, the type of message is manipulated, so as to present the attributes positively in the 1SM, and to describe one secondary attribute negatively in the 2SM. For the 1SM, the attributes were: (i) super tasty protein fudge; (ii) has no sodium; (iii) does not have lactose; and (iv) high in protein. To the 2SM was added the negative attribute: The product sits on the bottom of the package, and you need to mix before eating it.

To determine the conditions for the variable YouTuber's media metrics, we included at the beginning of the vlog information about subscribers, likes, and shares. To do so, a short online inquiry was posted, via the Qualtrics platform, to identify high numbers for a YouTuber. Through WhatsApp and social networks, 30 people were invited to participate in the inquiry. Participants were asked to choose the minimum number they considered as an indicator of good performance, both for a national YouTuber and for a national vlog, in three features: number of subscribers; number of likes; and number of shares. The numbers tested were: >10,000; >20,000; >50,000; >100,000; >200,000; >500,000; and >1,000,000. The survey was available during a two-day period. To choose the quantities for the YouTuber's media metrics conditions, we calculated the mean of the minimum number indicating good performance of a YouTuber and a vlog for each feature. For the high YouTuber's media metrics condition, the displayed numbers were: 527,235 subscribers, 283,000 likes, and 16,000 shares. For the condition of no information about a YouTuber's media metrics, only the 'sign up' button was inserted in the vlog. The two conditions for YouTuber's media metrics were included in the two types of vlogs (1SM and 2SM). Then, four vlogs were used as stimuli in the experiment.

### 3.2. Stimuli Pre-Test

To pre-test the stimuli, 30 students were exposed randomly and individually to one vlog of around 2 min. First, they were asked whether the message or not presented a product. Correctly, 100% of the respondents answered affirmatively. Then, they were asked whether the message about the product presented either: only positive, only negative, or both positive and negative features. Of the 15 participants exposed to the one-sided ad, 12 (80%) indicated that it had shown only positive features of the product, whereas of the 15 participants exposed to the two-sided ad, 14 (94%) indicated that it had shown both positive and negative features ($X2 = 16.425$, $p < 0.001$). Thus, it was concluded that the advertisements were recognized as one-sided and two-sided, respectively. Additionally, participants were asked if the vlog contained information about the YouTuber's subscribers, likes, and shares. All respondents answered correctly. Then, participants gave their evaluation of the YouTuber's

performance. The alternatives were: good, poor, and no information. Of the 16 participants exposed to the no information condition, 11 (69%) indicated that the vlog had no information, 2 (12%) participants indicated a good YouTuber's performance, and 3 (18%) indicated a poor YouTuber's performance (X2 = 16.425, $p < 0.001$). Of the 14 participants exposed to the high YouTuber's media metrics condition, no-one indicated that the vlog had no information, all participants indicated a good YouTuber's performance, and non-one indicated a poor YouTuber's performance (X2 = 22.969, $p < 0.001$).

### 3.3. Sample

The sampling procedure began with the professional contacts of an MBA student: a 32-year-old male, middle-class, advertising consultant. An invitation with a link using Qualtrics software was sent to 340 contacts, specifying the characteristics the respondents must meet. It was established that each respondent should be a social network user. The link remained available for a 10-day period. After four days, new messages were sent to the guests, reminding them of the invitation and informing them of the survey schedule. The final sample ($n = 125$, 52% female, $M_{age}$ = 40 years) was exposed randomly to one of the four conditions. Ninety percent of the respondents had at least a partial undergraduate degree, 60% had average income within the middle-class interval, and 20% had average income within the upper-class interval.

### 3.4. Procedure and Measurements

The survey was presented as marketing research about individuals' behavior in social networks. In the first part of the experiment, there were general questions about the use of social networks, such as time spent and access frequency during the day. Then, conformity intention towards social networks was measured. The instruction was "Now, think of the social network you signaled above as that one you use the most. Think about the people who are part of your social network as a group of individuals with common interests. You are part of this group! For this social networking group, answer the following questions". Continuing, the heading before the conformity intention items was "About that social network group you are thinking of now, indicate your degree of agreement with the following statements". Five items were used on a seven-point scale, anchored by 1 = strongly disagree and 7 = strongly agree. The statements were [37]: I am willing to agree with them/I am willing to follow their opinion/I am willing to join them as the same group member/I am willing to share the same opinion as the same group (Cronbach's $\alpha$ = 0.84, M = 3.88, SD = 1.18). In this research, conformity intention was used as a measured independent variable for the data analysis. After the measurement of conformity intention, two other self-characteristics (self-monitoring sensitivity and self-monitoring ability) were measured to distract the participants. No manipulation checks were included to avoid bias. At the conclusion of this research, the participants were invited to collaborate in another piece of research about a new YouTuber's work. All of the participants agree to participate in that research.

Respondents were exposed randomly to one vlog representing the 1SM or the 2SM condition and the high Tuber's numbers or no information about the YouTuber's media metrics. Then, the dependent variables: attitudes towards the vlog, the product, and the brand were assessed. For attitude towards the vlog, six items were used on a seven-point Likert scale anchored by 1 = strongly disagree and 7 = strongly agree [40]. The original, unidimensional scale for attitude towards the advertisement was adapted for measuring attitude towards the vlog. The items were: good/interesting/informative/appropriate/easy to understand/objective ($\alpha$ = 0.97, M = 3.47, SD = 1.15). For attitude towards the product, six items were used on a semantic differential scale [40]: bad/good; unfavorable/favorable; disagreeable/agreeable; unpleasant/pleasant; negative/positive; dislike/like ($\alpha$ = 0.98, M = 4.13, SD = 1.92). For attitude towards the brand, five items were used on a semantic differential scale [41]: unappealing/appealing; bad/good; unpleasant/pleasant; unfavorable/favorable; unlikable/likable ($\alpha$ = 0.98, M = 3.8, SD = 1.85). Additionally, the following control variables were measured: an individual's identification with the YouTuber, the YouTuber's credibility, and the YouTuber's attractiveness. These three variables were measured on a seven-point Likert scale anchored by 1 = strongly disagree and 7 = strongly agree. For

identification, the items were: Renata and I probably have similar beliefs and values/Renata is quite like me/Renata and I are likely to have similar interests ($\alpha$ = 0.92, M = 2.62, SD = 1.5). For credibility of the YouTuber, the items were: Renata is trustworthy/Renata is honest when recommending the Force Two fudge/Renata seems to be trustworthy/Renata seems to be sincere ($\alpha$ = 0.92, M = 3.45, SD = 1.5). For attractiveness of the YouTuber, the items were: Renata is attractive/in my opinion, Renata looks good/Renata is beautiful ($\alpha$ = 0.86, M = 5.16, SD = 1.20).

## 4. Results and Discussion

To analyze the interaction effect of type of message and a YouTuber's media metrics on attitudes towards the vlog (H1), the product (H2), and the brand (H3), three two-way analyses of variance (Two-way ANOVA) were performed on each dependent variable. To analyze the moderator effect of conformity intention on the effect of type of message X YouTuber's numbers on attitudes towards the vlog (H4), the product (H5), and the brand (H6), three three-way analyses of variance (Three-way ANOVA) on each dependent variable were performed. The hypotheses that were tested, the analytical techniques and the results obtained are summarized in Table 1.

**Table 1.** Hypotheses, analytical techniques, and results.

| | Hypotheses | Analysis | Result |
|---|---|---|---|
| H1a | 1SM: high YouTuber's media metrics gives a more positive attitude towards the vlog than without YouTuber's media metrics | Two-Way ANOVA | √ |
| H1b | 2SM: high YouTuber's media metrics gives a less positive attitude towards the vlog than without YouTuber's media metrics | | √ |
| H2a | 1SM: high YouTuber's media metrics gives a more positive attitude towards the product than without YouTuber's media metrics | Two-Way ANOVA | X |
| H2b | 2SM: high YouTuber's media metrics gives a less positive attitude towards the product than without YouTuber's media metrics | | |
| H3a | 1SM: high YouTuber's media metrics gives a more positive attitude towards the brand than without YouTuber's media metrics | Two-Way ANOVA | X |
| H3b | 2SM: high YouTuber's media metrics gives a less positive attitude towards the brand than without YouTuber's media metrics | | |
| H4 | Conformity intention moderates the interaction effect of type of message X YouTuber's media metrics on attitude towards the vlog | Three-Way and Two–Way ANOVA | √ |
| H5 | Conformity intention moderates the interaction effect of type of message X YouTuber's media metrics on attitude towards the product | Three-Way ANOVA | X |
| H6 | Conformity intention moderates the interaction effect of type of message X YouTuber's media metrics on attitude towards the brand | Three-Way ANOVA | X |

### 4.1. The Effect of Type of Message Moderated by a YouTuber's Media Metrics on Customers' Responses (H1, H2, and H3)

Three two-way, between-subjects ANOVAs were conducted to analyze the interaction effect of a YouTuber's media metrics conditions (high and no information) and the effect of type of message (1SM and 2SM) on attitudes towards the vlog, the product, and the brand. The moderator effect of a YouTuber's media metrics on the effect of type of message on customers' responses was supported only for attitude towards the vlog (H1). There was a statistically significant interaction between the effects of the type of message condition and a YouTuber's media metrics conditions on attitude towards the vlog ($F(1, 121)$ = 10.321, $p$ = 0.002), supporting H1a and H1b (Figure 2). As expected for a 1SM, individuals submitted to the high YouTuber's media metrics condition showed a more positive attitude towards the vlog (M = 3.95, SD = 1.63) than did individuals submitted to the no information of the YouTuber's media metrics condition (M = 2.96, SD = 1.39, $t(61)$ = −2.595, $p$ = 0.012). On the other

hand, for the 2SM, individuals submitted to the high YouTuber's media metrics condition showed a less positive attitude towards the vlog (M = 3.18, SD = 1.42) than did individuals submitted to the without the YouTuber's media metrics condition (M = 3.88, SD = 1.42, t(60) = 1.937, $p$ = 0.050). There were no interaction effects of the independent variables on the dependent variables: attitude towards the product (F(1, 121) = 0.873, $p$ = 0.352) and attitude towards the brand (F(1, 121) = 0.551, $p$ = 0.459), thus not supporting H2 and H3. The results were robust when the control variables: individual's identification with the YouTuber, credibility of the YouTuber, and trustworthiness of the YouTuber were added to the model (F(1, 118) = 7.418, $p$ = 0.007). This shows that the attitude towards the vlog is related to the interaction of type of message and the YouTuber's media metrics on the stimuli, and not to the level of the control variables.

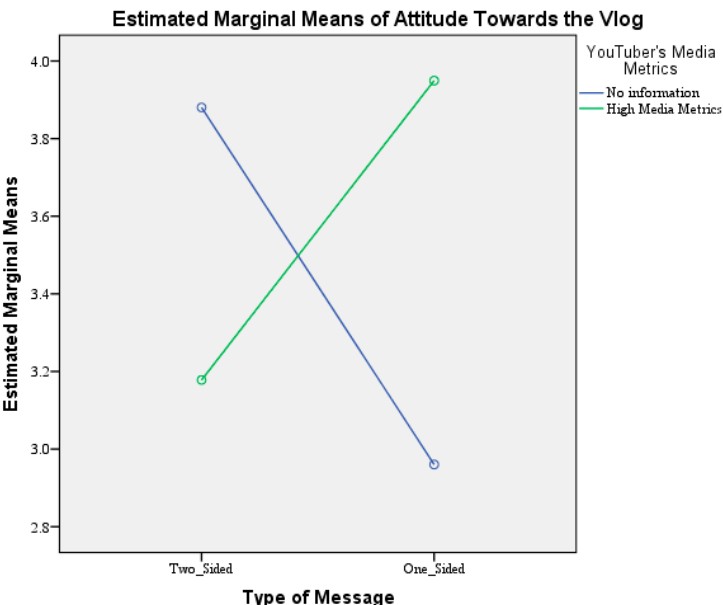

**Figure 2.** The effect of type of message on attitude towards the vlog moderated by the YouTuber's numbers.

Participants exposed to the 1SM had more positive attitudes when the high YouTuber's media metrics were informed, signaling low involvement with the message, which in turns leads participants to pay less attention to each other's idiosyncratic qualities, while making them aware of and focus on the YouTuber's media metrics [26,42]. If the YouTuber has high media metrics, the individual will pay less attention and have less motivation to process the 1SM [9], following the peripheral route of the ELM.

Conversely, we found that the effect of 2SMs on attitude towards the vlog is less positive with the high YouTuber's media metrics than without. This result is congruent with previous research showing that, in contrast to 1SMs, 2SMs increase involvement [9], and the individual follows the central route of persuasion to process the message [9]. Under the ELM, the explanation for the confirmed effect is that the negative information about the product may activate some initial disagreement towards the message, and for this reason, the individual will tend to engage in counter-arguing. As a consequence, they will decrease their attitude towards the vlog with high media metrics. This explanation is congruent with the moderator effect of distraction on the counter-argument production towards divergent messages proved in previous research [42]. Distraction during reception of advertisement messages was associated with decreased and increased levels of communication acceptance. For 1SMs, YouTubers' media metrics could be interpreted as a distraction factor triggering heuristic processing. Differently, for 2SMs, the negative information of the product in the message will neutralize the possible distraction exerted by the YouTubers' media metrics, increasing counter-arguing processing and lowering attitudes towards the vlog.

In the processing of a 1SM, to relegate decision making to the group is a factor that causes a lowering of objective self-awareness, following a deindividuated condition [43]. In this research, the high YouTuber's media metrics will cause lowering self-awareness, and the individual will have better evaluations towards the product, because they will extend the YouTuber's high media metrics to the evaluation of the vlog. By contrast, a 2SM would increase perceived novelty [9] and have an effect on customers' responses by following the attention and motivation pads, increasing self-awareness. For this reason, the individual exposed to a 2SM with the high YouTuber's media metrics will tend to be more critical in an effort to be different from the others. On the other side, when exposed to a 2SM without information about the YouTuber's media metrics, participants' attitudes towards the vlog are more positive, making the effect of the 2SM on its credibility and consequently on the responses to the vlog prevail.

A recent study analyzed peoples' susceptibility to reputation in the online environment [44]. It was found that a well-rated individual received more generous offers from others exposed to their high rate reputation. This means that the simple fact of assigning a good reputation to an individual appeared to be sufficient to prompt other individuals to reward them with more positive responses. Similarly, in our study, the vlog disclosing the high YouTuber's media metrics received more positive evaluations than the vlog without the YouTuber's media metrics information. The high YouTuber's media metrics may have been perceived as an indicator of the YouTuber's reputation. Moreover, those authors discovered that psychological variables moderate the relationship between online reputation and an individual's behavior [44]. They found that high-anxious and high-openness individuals are less prone to rewarding a high individual's reputation with positive responses. This psychological profile appeared to ignore the individuals' reputation in their decision-making process. In our study, the results regarding 2SMs are similar to those for high-anxious and high-openness individuals in the mentioned study. In the mentioned study, individuals are ignoring the other individual's reputation, while in our study, respondents exposed to a 2SM are ignoring the high YouTuber's media metrics, which may have been perceived as a reputation indicator. An explanation for this finding is that the 2SM could have triggered the individual's curiosity (related to openness) or generated anxiety feelings, diminishing the effect of the high YouTuber's media metrics information.

*4.2. The Moderator Effect of Conformity Intention on the Effect of Type of Message Moderated by a YouTuber's Media Metrics on Attitude towards the Vlog (H4)*

A three-way ANOVA was performed to analyze the moderator effect of conformity intention on the interaction effect between a YouTuber's media metrics conditions (high and no information) and type of message (1SM and 2SM) on each dependent variable: attitude towards the vlog, attitude towards the product, and attitude towards the brand. Before performing the three-way ANOVA, the continuous variable conformity intention was dichotomized by the median in two categories: low and high levels of conformity intention. Then, an ANOVA with three categorical independent variables (type of message, YouTuber's media metrics, and conformity intention) and each dependent variable was performed. The proposed moderation was supported for the dependent variable attitude towards the vlog (H4). There was a statistically significant interaction among the effects of the type of message, YouTuber's media metrics, and level of conformity intention on attitude towards the vlog ($F(1, 117) = 3.930$, $p = 0.050$), supporting H4. Table 2 shows the descriptive statistics for this analysis. The proposed moderation was not supported for the dependent variables attitude towards the product ($F(1, 117) = 1.050$, $p = 0.308$), and attitude towards the brand ($F(1, 117) = 0.901$, $p = 0.344$), thus not supporting H5 and H6.

**Table 2.** Descriptive statistics for attitude towards the vlog by type of message X YouTuber's media metrics and conformity intention.

| Type of Message | YouTuber's Media Metrics | Conformity Intention | Mean | Std. Deviation | N |
|---|---|---|---|---|---|
| Two_Sided | No information | Low | 4.04 | 1.499 | 18 |
| | | High | 3.56 | 1.254 | 9 |
| | | Total | 3.88 | 1.416 | 27 |
| | High Media Metrics | Low | 2.83 | 1.401 | 23 |
| | | High | 3.85 | 1.232 | 12 |
| | | Total | 3.18 | 1.414 | 35 |
| | Total | Low | 3.36 | 1.550 | 41 |
| | | High | 3.73 | 1.218 | 21 |
| | | Total | 3.48 | 1.447 | 62 |
| One_Sided | No information | Low | 2.94 | 1.460 | 21 |
| | | High | 3.00 | 1.286 | 10 |
| | | Total | 2.96 | 1.385 | 31 |
| | High Media Metrics | Low | 4.35 | 1.587 | 12 |
| | | High | 3.71 | 1.642 | 20 |
| | | Total | 3.95 | 1.626 | 32 |
| | Total | Low | 3.45 | 1.634 | 33 |
| | | High | 3.47 | 1.548 | 30 |
| | | Total | 3.46 | 1.581 | 63 |
| Total | No information | Low | 3.45 | 1.560 | 39 |
| | | High | 3.27 | 1.268 | 19 |
| | | Total | 3.39 | 1.462 | 58 |
| | High Media Metrics | Low | 3.35 | 1.618 | 35 |
| | | High | 3.76 | 1.482 | 32 |
| | | Total | 3.55 | 1.557 | 67 |
| | Total | Low | 3.40 | 1.578 | 74 |
| | | High | 3.58 | 1.414 | 51 |
| | | Total | 3.47 | 1.510 | 125 |

The significance of the moderation for conformity intention was analyzed by dividing the data into high and low levels of conformity intention and then performing a two-way ANOVA. Results confirmed that the interaction effect between type of message and a YouTuber's media metrics remains for individuals with low conformity intention ($F(1, 70) = 13.697$, $p = 0.000$) and disappears for individuals with high level of conformity intention ($F(1, 47) = 0.264$, $p = 0.610$). Graphically, Figure 3 shows that for individuals with a low level of conformity intention, a 2SM generates a more positive attitude towards the vlog when no information about YouTuber's media metrics is shown (M = 4.04, SD = 1.49, compared to when high a YouTuber's media metrics are displayed (M = 3.83, SD = 1.40, $t(39) = 2.659$, $p = 0.011$), and that a 1SM generates a more positive attitude towards the vlog when a high YouTuber's media metrics are shown (M = 4.35, SD = 1.58), compared to when no information about the YouTuber's media metrics is displayed (M = 2.94, SD = 1.46, $t(31) = -2.576$, $p = 0.015$). Differently, Figure 4 shows that for individuals with a high level of conformity intention, a 2SM generates the same attitude towards the vlog when no information about a YouTuber's media metrics is shown (M = 3.56, SD = 1.25 and when high a YouTuber's media metrics are displayed (M = 3.85, SD = 1.23, $t(19) = -0.515$, $p = 0.612$), and that a 1SM generates the same attitude towards the vlog when a high YouTuber's media metrics are shown (M = 3.71, SD = 1.64) and when no information about the YouTuber's media metrics is displayed (M = 3.00, SD = 1.28, $t(28) = -1.195$, $p = 0.242$). Figures 3 and 4 show the interaction effect between type of message and a YouTuber's media metrics is different for individuals with low and high levels of conformity Intention.

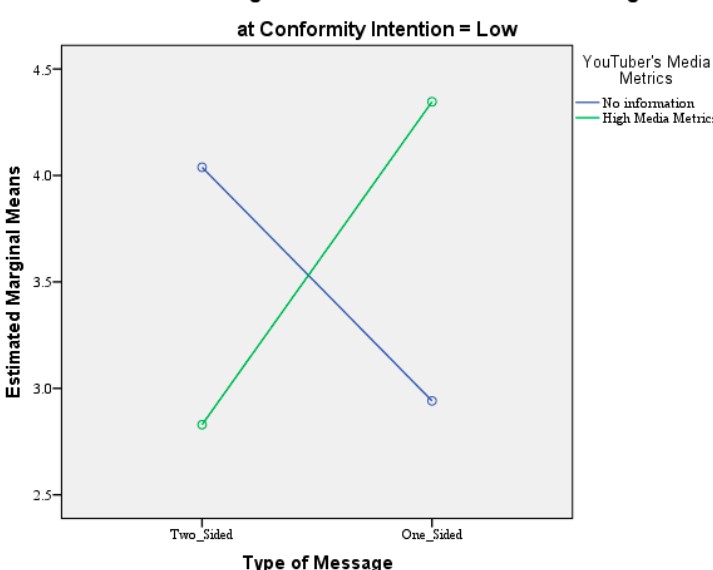

**Figure 3.** The interaction effect between type of message and YouTuber's media metrics on attitude towards the vlog for low conformity intention.

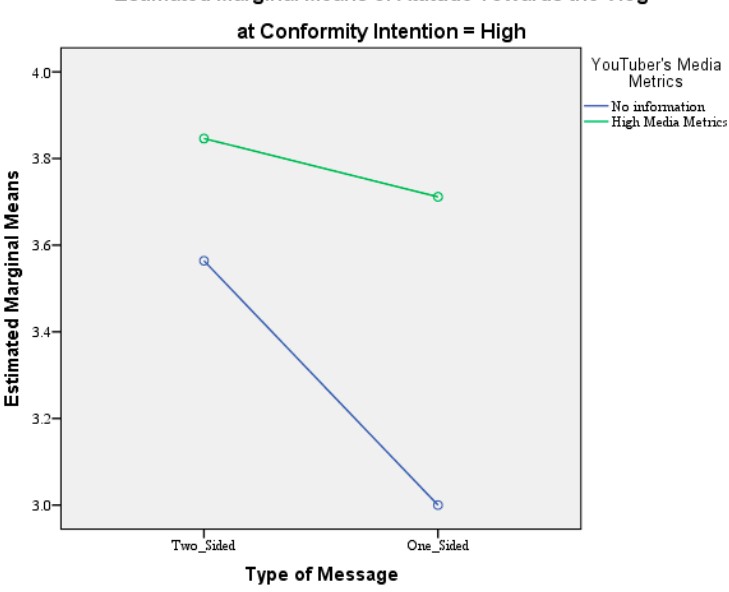

**Figure 4.** The interaction effect between type of message and YouTuber's media metrics on attitude towards the vlog for high conformity intention.

The explanation for the interaction effect of type of message X YouTuber's media metrics for individuals with low conformity intention is the same given for the analysis of the aggregated data, based on the ELM—but why does the moderator effect of conformity intention disappear for individuals with high conformity intention? This finding could be speculatively explained by the optimal distinctiveness theory (ODT), which posits that individuals always look for equilibrium between extreme similarity to others and extreme distinctiveness from others [37]. This means that individuals want to be moderately dissimilar in a group. Simultaneously, they want to be part of the group and different from others in the group. If individuals want to be part of the group and yet different from others in the group at the same time [42], the intensity of the need to be unique in the group and part of it would be conditioned by the individual's conformity intention. Thus, it is possible that an individual with high conformity intention will not have a high need for individualization.

For this reason, attitudes towards vlogs with different types of messages and information about media metrics were statistically the same and around the middle of the scale. This explanation is in line with the moderation role played by the psychological variables on the relationship between an online source's reputation behavior and the related individual's behavior, already proposed in recent literature [44]. It is known that individual's openness and anxiety moderate the effect of the source's reputation on an individual's responses [44]. Our speculation is that high-anxious and high-openness individuals could have low conformity intention, and low-anxious and low-openness individuals could have high conformity intention.

## 5. Conclusions

This research contributes to two lines of theoretical study. It augments the literature on 2SMs through the identification of factors that explain the effect of different types of messages in social networks. It adds to the research into online content and information use in general, supported by the ELM, by providing explanations on how media metrics influence the effect of two types of messages. The literature review showed that in online environments, compliance with social norms can emerge in different ways compared to face-to-face interaction [20]. Moreover, it is still unclear which elements can have the power to influence individuals' behavior during online communication processes [2], calling for more specific explanations.

It was speculated that the effect of 1SMs and 2SMs in social networks could be different in comparison to their effects in traditional media, because of particularities in media metrics and its effects on a customer's responses [30]. In addition, direct information seeking and involuntary exposure to information increasingly take place on social media platforms, such as Facebook, YouTube, and Instagram [45], increasing the customer's exposure to 2SMs. In this scenario, the first theoretical contribution of this investigation is the proposition and testing of the interaction effect between the type of message (1SM and 2SM) and the media metrics of the vlog on customers' responses. The results confirmed that the effect of 1SMs on attitude towards the vlog is more positive with a high YouTuber's media metrics than without that information and that the effect of 2SMs on attitude towards the vlog is more positive without information about a YouTuber's media metrics than with a high YouTuber's media metrics information. Regarding 1SMs, this confirmation supports previous studies on online content and information literature [27,30,39] that based their explanations on the bandwagon heuristic [33]. Regarding 2SMs, it also supports previous research in two-sided perception [2,8,9,15] and in computer-mediated communication [44].

Based on our results and theoretical explanations, media metrics of social networks must be considered as a moderator factor of the effect of different types of message. However, this confirmation needs replication efforts in future research. It is possible that subjects in the 2SM condition may have believed that they were being tested for their ability to perceive the negative side of the product and their responses, lowering attitudes, were somehow an attempt to demonstrate this ability. Additionally, the proposition and testing of the interaction effect between type of message and a YouTuber's media metrics on customers' responses was supported only for the dependent variable attitude towards the vlog. Attitudes towards the product and the brand inserted in the message were not influenced by the proposed independent variables. This is evidence that the type of message and media metrics of the vlog do not have enough power to influence evaluations of the announced product. This result confirms previous research, which demonstrated that Internet users rely mostly on surface characteristics and features rather than on content evaluation to give likes and shares [30]. This research extends the examination of the effect of media metrics on attitudes towards the vlog. A possible explanation for the lack of confirmation of H2 and H3 could be that information about the media metrics of the vlog is insufficient to increase an individual's confidence enough to make a judgment of the product and the brand.

In previous research, conformity intention to social norms was analyzed as a dependent variable of individuation and deindividuation processes [45]. We proposed a new perspective for the analysis of

conformity intention focusing on conformity intention to others' opinion in social networks. Thus, the second theoretical contribution of this research was to provide initial evidence of conformity intention towards social networks as a moderator factor on the interaction effect between type of message and a YouTuber's media metrics on customers' responses. Our hypotheses were confirmed for the proposed dependent variables attitude towards the vlog. It was found that at low levels of an individual's self-declared conformity intention in social networks, the evaluations of the 1SM vlog when a high YouTuber's media metrics are disclosed is better than if no information about a YouTuber's media metrics is shown. By contrast, the evaluations of the 2SM vlog when a high YouTuber's media metrics are disclosed is worse than if no information about YouTuber's media metrics is shown. In other words, at low levels of an individual's conformity intention, the 1SM will generate a more positive attitude towards the vlog by disclosing high media metrics, and the 2SM will generate a more positive attitude towards the vlog by not disclosing media metrics. On the other side, at high levels of an individual's self-declared conformity intention in social networks, the evaluations of the 1SM and the 2SM are the same when showing a high YouTuber's media metrics and without information about the YouTuber's media metrics. In other words, the effect of media metrics of the vlog increases when the individual has low propensity to conform, disappearing at high levels of conformity intention. Our research provides an alternative explanation, namely that it is conformity intention to others' opinions which would determine the customer's responses towards messages with different characteristics. Moreover, regarding our explanations based on the ODT and on the psychological individual's profile, more research is required. Future studies must clarify how an individual's characteristics moderate the persuasive effect of a message's sidedness.

The exposure of consumers to multiple sources of information has increased their interaction with 2SMs. In this situation, the acquisition of knowledge for better management decisions is also particularly relevant. The findings of this research highlight the importance of a company's knowledge of customers' conformity intention to others' opinions tendency. A previous study has highlighted the need for communication practitioners to pay closer attention to the cognitive traits of their target audience when designing and distributing persuasive messages [27]. In this regard, this research extends this recommendation to pay closer attention to situational factors that influence the processing of communication messages. Customers having a low level of conformity intention would have better responses in front of 1SMs disclosing a high YouTuber's media metrics, compared to 1SMs without a YouTuber's media metrics, whilst customers having a low level of conformity intention would have better responses in front of 2SMs without a YouTuber's media metrics, compared to 2SMs disclosing a high YouTuber's media metrics. For this reason, companies can positively influence individuals' conformity intention by managing bi-directional communication (e.g., using social networks), customizing communications oriented by this objective. Using interactive media, it is possible to customize marketing communications [46], increasing the customers' conformity intention.

From a theoretical perspective, this research's limitations indicate avenues for future research. The ODT is a complex theory involving a lot of different variables that have not been considered in this investigation. Evidence to confirm speculations in this research involving this theory opens venues for future studies. In light of the decision process theory, this research explains the mechanism of the moderator effect of conformity intention on the interaction effect of type of message and media metrics on attitudes towards the vlog. Specifically, the level of conformity intention towards behavior in social networks was measured before exposure to a message. However, the level of conformity intention towards opinions in the message after that exposure was not. Future research must analyze both objects of conformity intention: towards social networks in general and towards the opinions in the message. In subsequent studies, the analysis of different objects of conformity intention and other personal characteristics of the individual could result in better explanations for the effect of type of message and media metrics.

In addition, the high level of conformity intention to others' opinions in social networks and the speculated no need of individualization, as well as the low level of conformity intention and high need

for individualization, must be confirmed in future studies. An explanation is needed as to how and why that influence happens. It was postulated that an individual's level of conformity intention would influence his or her need for individualization and to be part of the group. Thus, future research should aim to clarify which related constructs mediate the effect of type of message by level of conformity intention, and the mechanism of that mediation. Additionally, it would be of value to test other characteristics of individuals as moderators for the effect of type of message and media metrics, such as personality, mood, and emotional background (e.g., nervousness, anxiety, and distraction).

From a methodological perspective, there are also limitations to be rectified in future studies. This research analyzed only one product category. Future research should cover a broader range of categories and concepts of goods and services, which could yield different results and explanations. Another methodological limitation of this study is inherent in the procedure of convenience sampling. However, it is incumbent on the researchers to be concerned about the generalizability of research results beyond the lab. In this regard, it is known that the more similar subjects are to actual consumers, the more the research results may be generalized and transferred from an experiment to aid a marketing manager's actual decisions. Thus, to increase the external validity, future research should recruit participants with a profile similar to that of real consumers. This would give marketing professionals the confidence to apply these results to their real markets. There are also limitations related to the stimuli (vlogs): They were not created by advertising professionals, and that may have reduced their credibility. Methodologically, future studies should replicate the findings of this research using different product categories, manipulations, participants, and procedures, in order to test the generalizability of the findings reported.

**Author Contributions:** M.K.Z.H.: Proposal, analysis and writing; T.D.C.: Empirical Research implementation.

**Funding:** This research received no external funding.

**Acknowledgments:** The authors would like to thank José Mauro da Costa Hernandez for his suggestions on the statistical analysis.

**Conflicts of Interest:** The authors declare no conflict of interest.

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
