# Peer review of "The Interaction Effect of Type of Message X YouTuber’s Media Metrics on Customers’ Responses and the Moderation of Conformity Intention"

_futureinternet, doi:10.3390/fi11060135_

Round 1
Reviewer 1 Report
The topic of effectiveness and impact of messages published on social media is quite relevant. The article studies the relations among the media metrics, the conformity intention, the type of message, and its perceived effectiveness. Two types of vlog messages published on YouTube are considered: one-sided and two-sided messages. Evaluations are based on the answers of inquiries submitted to various panels, each including few dozens of qualified respondents. The research work is interesting and well conducted. The article is well organized to present the research work with sufficient clarity. Results are discussed appropriately. References are adequate and up-to-date.
There are few language issues, for example:
Maybe “YouTuber's” should really be “YouTubers'” in many cases.
The use of nested and consecutive parenthesis makes some sentences difficult to read. Please, avoid it.
Line 74 is ill formed: “seems relevant research ...”
Line 488 cenário
Author Response
We would like to thank the reviewer for the constructive comments. All his suggestions have been incorporated into the manuscript, as detailed in the next table. We would like to inform that the manuscript has been reviewed and edited by a professional from Peerwith.
Reviewer’s Comment | Answer |
The topic of effectiveness and impact of messages published on social media is quite relevant. The article studies the relations among the media metrics, the conformity intention, the type of message, and its perceived effectiveness. Two types of vlog messages published on YouTube are considered: one-sided and two-sided messages. Evaluations are based on the answers of inquiries submitted to various panels, each including few dozens of qualified respondents. The research work is interesting and well conducted. The article is well organized to present the research work with sufficient clarity. Results are discussed appropriately. References are adequate and up-to-date.
| Thank you for the encouraging comments. |
There are few language issues, for example: Maybe “YouTuber's” should really be “YouTubers'” in many cases.
| The language correctness was reviewed. |
The use of nested and consecutive parenthesis makes some sentences difficult to read. Please, avoid it.
| All the manuscript has been reviewed following this suggestion. |
Line 74 is ill formed: “seems relevant research ...”
| The sentence was corrected. Line 78-80: In this context, it is relevant to analyse the effects of social media metrics and the specifics of content on individuals’ responses.
|
Line 488 cenário
| The word was changed. |
Reviewer 2 Report
This work explores the interaction between the effect of type of message, youtuber’s numbers on customer’s responses and the moderation effect of conformity intention.
The research is potentially interesting. However, as it stands now, the paper is very far from being publishable and should be highly revised. In particular, the two major shortcomings of this paper are the following:
1) In general, the paper appears to be unclear and difficult to understand. The authors should pay more attention to their writing, because sometimes the reader is forced to interpret the text and try to imagine what the authors meant. Furthermore, in the text there are also some grammatical errors, some spelling errors, and missing words that prevent the correct understanding of the text’s content. An extensive revision of the English and the style of should be done, because as it stands now, the paper it’s hard to be understood both from experts and people that come from a different research field.
2) The theoretical background of the study appears to be weak or, sometimes, inappropriate. Furthermore, the authors should take into consideration the scientific literature on attitudes, persuasive communication, and reputation since it could be useful to interpret the results of the study
I suggest a major revision, inviting the authors to revise carefully and deeply the manuscript and the literature.
Particular attention should be paid to the construct of Deindividuation, which is very complicated and elusive in real experiments.
The paper is maybe too ambitious, and of course contains interesting ideas as well as theorethical attempts to understand the results. Unfortunately, beyond the lot of typos within the text, the main findings should be revised and reinterpreted.
More details:
- The abstract should be rewritten. Below some comments.
Line 11-13: The first statement of an abstract is very important, so it should be written clearly, for example: "The present study aims to explore the influence that YouTuber’s media metrics and YouTuber’s one-sided (1SMs) and two-sided (2SMs) messages exert on costumers’ responses in the light of the Social Identity Model of Deindividuation Effect (SIDE model)”.
Line 13-17: In general, both the objectives of the study should be better described and written. Furthermore, the phrase “high YouTuber’s numbers or no information” between brackets is not clear, so it should be improved or removed.
Line 18: “Results of an experimental study confirm that high YouTuber’s numbers have more effect on 1SMs and less effect for 2SMs”. It seems that the high YouTuber’s numbers have effect on the type of message, while the objective of the study was to study the interaction between the YouTuber’s numbers and the type of message on customer’s responses. If the result refers to the first objective of the study, it should be rewritten properly.
Line 20-21: Also in this case, the authors should explain that the conformity intention increases the effect of YouTuber’s numbers on customer’s responses.
Line 21-23: Maybe the authors meant that the “study provides a theoretical contribution..”. Also this statement should be written better.
Line 11-23: The authors should use always the same term to refer to YouTuber’s media metrics to avoid confusion, since they sometimes speak about “vlog media metrics”, sometimes about “YouTuber’s numbers” and so on. Please unify the nomenclature.
Line 24: I advise to delete “two-sided messages” from the keywords and replace it with “type of message” or to add to the keywords also “one-sided messages” for completeness.
- The introduction should be improved, in terms of English, style, and content. There are also spelling mistakes that should be corrected.
Line 28-30: The same problem of the abstract. It should be rewritten.
Line 28-57: This part of introduction should be enlarged and enriched with the extensive literature concerning persuasive communication. Here some examples of literature contributions that can help the authors to extend the handling of the argument.
1) McGuire, W. J. (1978). An Information Processing Model of Advertising Effectiveness. Behavioral and Management Science in Marketing, eds. Harry L. Davis and Alvin H. Silk.
2) Petty, R. E., & Cacioppo, J. T. (1986). The elaboration likelihood model of persuasion. In Communication and persuasion (pp. 1-24). Springer, New York, NY.
3) Chaiken, S. (1987). The heuristic model of persuasion. In Social influence: the ontario symposium (Vol. 5, pp. 3-39).
4) Haney, W. V. (1964). A comparative study of unilateral and bilateral communication. Academy of Management Journal, 7(2), 128-136.
5) Osterhouse, R. A., & Brock, T. C. (1970). Distraction increases yielding to propaganda by inhibiting counterarguing. Journal of Personality and Social Psychology, 15(4), 344.
Line 77-82: The authors should better explain the “media metrics” concept within the body of the introduction and not leavening it within brackets in the last lines of the introduction. It’s a one of the central concepts of the paper, so it should have more space in the handling. Furthermore, it’s better to uniform the nomenclature of this concept throughout the paper.
- The Hypotheses Development section should be improved
Line 89-91: The concept of reputation fits the context; therefore, authors can cite the literature about it. For example, they can mention the individual differences that characterize people in front of reputation. The authors could cite Collodi et al. (2018), that present these differences in the article “Personality and Reputation: A Complex Relationship in Virtual Environments”.
Line 94-99: This statement should be written more clearly.
Line 108: It is better to cite also here the authors of SIDE model.
Line 109: It is better to speak about “depersonalization” and not “deindividuation” even if the title of the model is tricky.
Line 109-112: The authors should better handle the SIDE model, explaining for example the importance of social cues in front of other missing cues (e.g., facial expression, gestures, ecc..). They should highlight the importance of the social community and social group to which the individuals belong.
Line 112-118: Even if the paper architecture sounds interesting, the forecasting based on the SIDE model here appear to me as quite forced. In particular, my opinion is that of course the Deindividuation process should be considered to interpret some results, at the same time the fundamental ingredients required by the SIDE model appears as missing.
Line 118-122: The authors should define the media metrics concept before and use uniform nomenclature.
Line 123: It seems that the subject of the sentence is missing
Line 141-147: Here I don’t understand why the effect should be present when there is a 2SM and not in the case of a 1SM. Moreover my opinion is that the deindividuation effect appears as a weak theorethical background to explain these effects. The authors should revise this aspect.
Line 162: The word feelings is inappropriate. Please consider choosing more adequate ones in relation to the constructs between brackets.
Line 170-171: Please correct the statement because is unintelligible
Line 179-180: The sentence should be reworded.
Line 228-229: For the sake of clarity, the authors should avoid the text in the brackets or find a better way to express the concept.
Line 267-283: The authors should replace “p = 0.000” with “p <.001”.
Line 280-283: Summarize in one statement your finding, without presenting the “0 (0%)”.
Line 342: Table 1 should be fixed. Please insert the first result and remove the citation in “H4 Analysis”
Line 357: The authors should report the result as follows [F(1,121) = 10.321, p = .002]. Please uniform all the following results.
In general a lot of typos and other errors are present in all the text. For instance (but it is necessary to pay attention to the rest) hereafter I reported some examples:
381: (R2 381 = .1713, F(7,00) = 3.45, p = .0021). The degree of freedom cannot be 00, and the brackects again should be squared if within the others.
383: (b = -.9271, SD = .4145, t(117) = -2.2364, p 383 = 0,027). The b should be a “b”
Line 384-387: It is necessary to specify how the categories low, medium, and high conformity have been created.
In all the text:
- When is possible I would suggest to add the standard deviation in the bars’ figures.
- The captions are quite obscures, please try to explain better the contents of the figure (In the APA style every figure should be perfectly comprehensible without reading the paper).
- Please add in every figure a label explaining the axes contents.
- Table 2-3: again the caption appears not complete.
Line 474-481: This part should be improved. It can start with the rationale that underlies the research and why it appears to be interesting and useful. Also, the part on the contributions that this research give to the literature should be enlarged. It could be useful to rewrite this part in a catchier way.
Line 562 – I would erase the following sentence, my doubt is that the results interpretation is very weak! In particular, to ask for the intention to conform is not related with real conformity, and moreover, the deindividuated states of subjects was not assessed perfectly but just tried to be manipulated.
Author Response
Future Internet
Responses to the comments of Reviewer #2
The Interaction Effect of Type of Message X YouTuber’s media metrics on Customers’ Responses and the Moderation of Conformity Intention
We would like to thank the reviewer for the constructive comments. All his suggestions have been incorporated into the manuscript, as detailed in the attached. We would like to inform that the manuscript has been reviewed and edited by a professional from Peerwith. Please note that the English in the attached “responses” document was not reviewed by this professional and I apologize for mistakes in my answers.
Reviewer’s Comments | Answers |

Round 2
Reviewer 2 Report
Dear authors, thank you very much for your revised version of the paper. I appreciate the extensive work that you performed, and the paper appears to be quite good but still, a few things should be modified before considering it publishable. I think that adjusting some minor inexactitudes the work can be considered complete.
Unfortunately, throughout the paper, the reference to the construct of deindividuation and the SIDE model is forced and not appropriate, so I suggest the authors to entirely eliminate it from their paper. I asked for better explanations in the last review because the article was not very clear, and I was wondering if I was missing some connection, but I think that this connection does not exist. I think also that the authors reached a good way of interpreting their results referring to the Attitudes and Behavioral Psychology, so there is no need to provide further explanations that don’t fit well with their study (i.e., ODT, need of individualization, need to belong). So, I suggest avoiding too creative interpretations, because the theories used to explain the results are complex, involve a lot of different variables that have been not considered and cannot be simply applied to the study in a guessing style.
Furthermore, the construct of conformity intention appears problematic to me as it stands now. Conformity intention is a complex construct highly dependent on the context, its norms, and the group dynamics so one should be careful to talk about it as an “individual characteristic”. This evidence is provided, for example, by Kim (2011) in the paper “Two routes leading to conformity intention in computer-mediated groups: Matching versus mismatching virtual representations” and also by the SIDE model (Postmes, Spears, and Lea, 1998). Also, the way in which conformism intention has been measured makes me think more about it as Virtual Sense of Community than conformity itself. So, I suggest the authors reconsider it in the light of this construct because, to me, it’s hard to talk about it as conformity intention.
For what concerns the results I would like a clarification. In Lines 449-458 the authors state “individuals submitted to the high YouTuber’s media metrics condition showed a more positive attitude towards the vlog (M = 3.95, SD = 1.63), than did individuals submitted to the no information of YouTuber’s media metrics condition (M = 2.96, SD = 1.39). On the other hand, for the 2SM, individuals submitted to the high YouTuber’s s media metrics condition showed a less positive attitude towards the vlog (M = 3.18, SD = 1.42), than did individuals submitted to the without YouTuber’s media metrics condition (M = 3.88, SD = 1.42). There was a statistically significant interaction between the effects of the type of message condition and YouTuber’s media metrics on attitude towards the vlog, [F(1, 121) = 10.321, p = .002,], supporting H1a and H1b (Figure 2)”. Then, in Lines 503-532 the effect is turned upside down considering the conformity intention. But Figure 3 makes me wonder because if I consider the data in an aggregate way and I don’t distinguish for conformity intention (for example summing the bars and take the average ) in 1SM the “high media metrics” bar should be greater than the “no information” bar according to the results in lines 449-458. The same reasoning for the 2SM and the following results about the product and the brand. Am I wrong? Did I miss something?
Line 469: I think that there is a mistake. I think that the authors refer to the attitude toward the vlog and not the brand.
Moreover, I suggest to the authors to improve the discussion of the results softening their language and being more cautious in making confident assertions. Finally, I also suggest amalgamating better the results of the study with the literature evidence, without presenting them separately, since the point of a discussion is to interpret the results in the light of theories.
Bibliography
Kim, J. (2011). Two routes leading to conformity intention in computer-mediated groups: Matching versus mismatching virtual representations. Journal of Computer-Mediated Communication, 16(2), 271-287.
Postmes, T., Spears, R., & Lea, M. (1998). Breaching or building social boundaries? SIDE-effects of computer-mediated communication. Communication research, 25(6), 689-715.
Author Response
Responses to the comments of Reviewer #2
The Interaction Effect of Type of Message X YouTuber’s media metrics on Customers’ Responses and the Moderation of Conformity Intention
We would like to thank the reviewer for the constructive comments.
We have carefully worked on all of them.

Round 3
Reviewer 2 Report
Dear authors, I appreciate the work that you did and the changing that you performed. Now I believe that the paper can be considered sufficiently good to be publishable after a couple of minor revisions. The first would be to reintroduce some recent references that would support your last considerations about deindividuation (such as Collodi et al.), and finally a last reading of the text since I noticed some English mistakes.
Author Response
Thank you very much to the reviewer for his comments and suggestions.
Suggestion |
The first would be to reintroduce some recent references that would support your last considerations about deindividuation (such as Collodi et al.).
|
Response |
I would like to thanks the reviewer for the suggested paper. I have read it and found it very interesting and related with my results a conclusions (“Personality and Reputation: A Complex Relationship in Virtual Environments”, Collodi et al., 2018). Based on the conclusion reported in the suggested paper I have incorporated the following paragraphs.
4.1. The effect of type of message moderated by YouTuber’s media metrics on customers’ responses (H1, H2 and H3)
Lines<433 449="">
A recent study analysed peoples’ susceptibility to reputation in the online environment [46]. It was found that a well-rated individual received more generous offers from others exposed to his high rate reputation. This means that the simple fact to assign a good reputation to an individual appeared to be sufficient to prompt other individuals to reward them with more positive responses. Similarly, in our study, the vlog disclosing high YouTuber’s media metrics received more positive evaluations than the vlog without YouTuber’s media metrics information. High YouTuber’s media metrics may have been perceived as an indicator of the YouTuber's reputation. Moreover, those authors discovered that psychological variables moderate the relationship between online reputation and individual’s behaviour [46]. They found that high-anxious and high-openness individuals are less prone to reward high individual’s reputation with positive responses. This psychological profile appeared to ignore the individuals’ reputation in their decision-making process. In our study, the results regarding 2SMs are similar to those for hig-anxious and high-openness individuals in the mentioned study. In the mentioned study, individuals are ignoring the other individual’s reputation while in our study, respondents exposed to a 2SM are ignoring the high YouTuber’s media metrics, that may have been perceived as a reputation indicator. An explanation for this finding is that the 2SM could have been triggered the individual’s curiosity (related to openness) or generated anxiety feellings, diminishing the effect of the high YouTuber’s media metrics information.
4.2. The moderator effect of conformity intention on the effect of type of message moderated by YouTuber’s media metrics on attitude towards the Vlog (H4)
Lines<511 -517="">
This explanation is in line with the moderation role played by the psychological variables on the relationship between online source’s reputation behavior and the related individual’s behavior, already proposed in recent literature [46]. It is known that individual’s openness and anxiety moderates the effect of source’s reputation on individual’s responses [46]. Our speculation is that high-anxious and high-openness individuals could have low conformity intention and low-anxious and low-openness individuals could have high conformity intention.
5. Conclusions
Lines<535 -541=""> The results confirmed that the effect of 1SMs on attitude towards the vlog is more positive with high YouTuber’s media metrics than without that information and that the effect of 2SMs on attitude towards the vlog is more positive without information about YouTuber’s media metrics than with high YouTuber’s media metrics information. Regarding 1SMs, this confirmation supports previous studies in online content and information literature [27, 30, 39] that bases their explanations on the bandwagon heuristic [33]. Regarding 2SMs, it also supports previous research in two-sided perception [2, 8, 9, 15] and in computer-mediated communication [46].
Lines<580 -582=""> Moreover, regarding our explanations based on the ODT and on the psychological individual’s profile, more research is required. Future studies must to clarify how individual’s characteristics moderates the persuasive effect of message’s sidedness.
|
Suggestion |
A last reading of the text since I noticed some English mistakes.
|
Response |
I have read all the manuscript and I found four mistakes. Thank you for this recommendation.
|
